# Transcriptome Meta-Analysis Identifies Candidate Hub Genes and Pathways of Pathogen Stress Responses in *Arabidopsis thaliana*

**DOI:** 10.3390/biology11081155

**Published:** 2022-08-01

**Authors:** Yaser Biniaz, Ahmad Tahmasebi, Aminallah Tahmasebi, Benedicte Riber Albrectsen, Péter Poczai, Alireza Afsharifar

**Affiliations:** 1Plant Virology Research Center, Faculty of Agriculture, Shiraz University, Shiraz 7194685115, Iran; yser.biniaz@shirazu.ac.ir; 2Institute of Biotechnology, Faculty of Agriculture, Shiraz University, Shiraz 7194685115, Iran; atahmasebi1@gmail.com; 3Department of Agriculture, Minab Higher Education Center, University of Hormozgan, Bandar Abbas 7916193145, Iran; a.tahmasbi@hormozgan.ac.ir; 4Plant Protection Research Group, University of Hormozgan, Bandar Abbas 7916193145, Iran; 5Department of Plant Physiology, Faculty of Science and Technology, Umeå University, 901 87 Umeå, Sweden; benedicte.albrectsen@umu.se; 6Botany Unit, Finnish Museum of Natural History, University of Helsinki, P.O. Box 7, FI-00014 Helsinki, Finland; 7Faculty of Biological and Environmental Sciences, University of Helsinki, P.O. Box 65, FI-00065 Helsinki, Finland; 8Institute of Advanced Studies Kőszeg (iASK), P.O. Box 4, H-9731 Kőszeg, Hungary

**Keywords:** *Arabidopsis thaliana*, biotic stress, plant–pathogen interaction, transcriptome data

## Abstract

**Simple Summary:**

Meta-analysis and systems-biology analysis revealed molecular plant defense responses in *Arabidopsis thaliana* when attacked by various pathogens. Differentially expressed genes were involved in several biosynthetic metabolic pathways, including those responsible for the biosynthesis of secondary metabolites and pathways central to photosynthesis and plant–pathogen interactions. In addition, WRKY40, WRKY46, and STZ transcription factors served as major points in protein–protein interactions. Overall, the findings highlighted genes that are commonly expressed during plant–pathogen interactions and will be useful in the development of novel genetic resistance strategies.

**Abstract:**

Following a pathogen attack, plants defend themselves using multiple defense mechanisms to prevent infections. We used a meta-analysis and systems-biology analysis to search for general molecular plant defense responses from transcriptomic data reported from different pathogen attacks in *Arabidopsis thaliana*. Data from seven studies were subjected to meta-analysis, which revealed a total of 3694 differentially expressed genes (DEGs), where both healthy and infected plants were considered. Gene Ontology and Kyoto Encyclopedia of Genes and Genomes pathway enrichment analysis further suggested that the DEGs were involved in several biosynthetic metabolic pathways, including those responsible for the biosynthesis of secondary metabolites and pathways central to photosynthesis and plant–pathogen interactions. Using network analysis, we highlight the importance of WRKY40, WRKY46 and STZ, and suggest that they serve as major points in protein–protein interactions. This is especially true regarding networks of composite-metabolic responses by pathogens. In summary, this research provides a new approach that illuminates how different mechanisms of transcriptome responses can be activated in plants under pathogen infection and indicates that common genes vary in their ability to regulate plant responses to the pathogens studied herein.

## 1. Introduction

Plants are continually confronted with a wide range of organismal infections and insect attacks, often leading to major losses in quantity and quality of plant products. In response to different types of pathogens, plants have adopted intricate defense systems that are structurally multilayered. These include PAMP triggered immunity (PTI), effector-triggered immunity (ETI), and RNA silencing [1,2,3]. In addition, plants have developed miRNA/target regulation pathways to maintain plant resistance by modifying the level of gene expression involved in plant defense systems [4]. Plant immunity is fundamentally based on the recognition of non-host organisms and protection against alien molecules [5]. In response to a pathogenic attack, plants reconfigure their cellular metabolism and induce a fine-tuned defense route that is compatible with the feeding behavior of pathogens. When pathogens are recognized through transmembrane pattern-recognition receptors (PRRs), for example, a cascade of responses is initiated, which ultimately results in the activation of first-line defense, known as PTI [6]. Some pathogens can adapt to this PTI defense by secreting effectors into the host cell. However, plants can prevail against the pathogenic repression of PTI via a second layer of the plant immune system, ETI [7,8]. A localized induction of ETI frequently leads to a broad-spectrum immunity that suppresses pathogenic infection in distal plant tissues, a phenomenon known as systemic acquired resistance (SAR) [9].

There have been extensive studies on plant transcriptome profiling against various pathogens [3,10,11,12]. Nonetheless, to the best of our knowledge, there is a lack of systematic information on the comparison of transcriptomic data across plant pathogen experiments [13,14,15]. With advances in high-throughput technology, such as RNA sequencing, large numbers of plant transcriptome responses to environmental stresses are becoming publicly available [16,17]. However, such experiments usually report responses to specific pathogens under experimental conditions.

A meta-analysis is a powerful tool that integrates results from several studies to draw general conclusions from several datasets, thereby enabling the detection of core gene hubs that are regulatory centers for complex biological processes in plants [18,19]. Whereas meta-analyses on transcriptome data (including microarrays and RNA-Seq data) have been widely applied in human and animal genome studies, only a limited number have focused on plants. Even fewer have explored plant responses to pathogenic infections [3,13,20]. Given the triple-layered challenges in agricultural systems [21], meta-analyses are useful tools that assist in summarizing current knowledge from which alternative solutions can be identified [22].

Performing a meta-analysis allows transcriptomic data across differentially expressed genes (DEG) to be integrated, thereby facilitating the discovery of major genes of general importance to plant stress-response [13,16,23]. These results can be used to validate singular transcriptomic works and assist researchers in gaining insights into general plant responses. In addition, pooling plant responses to various pathogens can allow researchers to uncover common features of plant–pathogen interactions [17]. Studying the transcriptome of *Arabidopsis thaliana* as a model plant, focusing on its various responses to stress, has generated large amounts of data that could potentially be used in a meta-analysis [3].

To understand the common transcriptional regulation of plants subjected to pathogen infection, we included publicly available RNA-Seq data from studies on *Arabidopsis thaliana* and its response to biotic stress. This was done to search for key determinants (including central genes, pathways, gene set categories, and protein–protein interaction networks) through meta-analysis and functional enrichment analyses. We identified DEGs involved in several diverse metabolic pathways, including transcription factors (TFs) and miRNA families. Moreover, a systems-biology analysis was employed to identify key regulatory hubs. The integrative approaches of this study provided insights into the mechanisms that regulate common defense responses of *Arabidopsis* when confronted by a variety of pathogens.

## 2. Materials and Methods

### 2.1. Data Collection and Preprocessing

The raw expression data (RNA-Seq) for biotic responses included those of fungi, oomycete, bacteria, and one viral strain in *Arabidopsis*. These RNA-Seq were obtained from the ArrayExpress of the European Molecular Biology Laboratory–European Bioinformatics Institute (http://www.ebi.ac.uk/arrayexpress, accessed on 1 March 2021) (Table 1). The keywords “biotic stress”, “pathogen”, “plant-pathogen interaction”, and various combinations of these were used to search the database. The datasets were filtered for *Arabidopsis thaliana* to include RNA-Seq data only. The search yielded seven entries in seven papers [24,25,26,27,28,29,30]. From these, we obtained transcript data of 284 individual plants, 103 of which were untreated controls and 181 of which were infected with pathogens, as detailed in Table 1.

The raw data were filtered during a quality control step. More specifically, reads with an N rate > 10% and bases on their Phred quality scores with *Q* ≤ 20 were eliminated [31]. Quality-checked reads were mapped onto the *Arabidopsis* reference genome. The *Arabidopsis thaliana* reference genome sequence (version TAIR10, release 31) was obtained from the EnsemblPlants database. Expression profiling analysis was carried out using CLC Genomic Workbench version 10 (CLC Bio, Qiagen, Hilden, Germany). The raw expression data pertaining to each dataset was normalized as counts per million (CPM). The workflow is presented in Figure 1.

### 2.2. Meta-Analysis of Expression Dataset

The meta-analysis was performed on an integrated dataset pertaining to DEGs in plant–pathogen interactions; about 20% of the genes with low expression levels were excluded to reduce the number of false positives in the samples. Each dataset was grouped into a stress class and a healthy class, according to the type of pathogen. Before the meta-analysis, the SVA package in R was used to correct the batch effect according to the empirical Bayes method [32]. Fisher’s method was used for detecting DEGs involved in plant–pathogen interactions. The adjusted *p*-values (FDR < 0.01) [33] were considered significant, and were used for further analysis. The log ratio of means (ROM), i.e., the natural log of the ratio [34], was applied to measure gene expression values. ROM was calculated using the following formula:$$y_{gn}=ln\left[\frac{{\bar{r}}_{gr}}{{\bar{r}}_{gs}}\right]$$
where *y_gn_*, *r_gr_*, and *r_gs_* represent the ROM, and the mean expression level of the stress and healthy class, respectively, for each gene in the dataset. Data were preprocessed and analyzed using Bioconductor packages (http://www.bioconductor.org, accessed on 5 April 2021), including MetaMA.

### 2.3. Gene Enrichment Analysis and Functional Analysis

The genes were analyzed after being selected for the meta-analysis. The obtained lists were compared with a list of genes involved in plant immunity responses and signaling processes. These included pattern recognition receptors, signaling complex, Ca^2+^ signaling system, G-protein signaling, reactive oxygen species (ROS) signaling system, nitric oxide (NO) signaling system, mitogen-activated protein kinase signaling system, salicylic acid signaling system, jasmonate signaling system, ethylene signaling system, and pathogen resistance proteins [4,6,35,36,37].

Enrichment analysis of Gene Ontology (GO) was performed on significant DEGs obtained from the meta-analysis, using the AgriGO platform [38]. Information on gene ontology was extracted based on GO terms for cellular components, biological processes, and molecular functions when a significant threshold of FDR < 0.05 was obtained. Pathway analysis of Kyoto Encyclopedia of Genes and Genomes (KEGG) (http://david.abcc.ncifcrf.gov/, accessed on 15 April 2021) was used to elucidate significantly enriched pathways of the DEGs. To identify transcription factors among the DEGs, a list of *Arabidopsis* transcription factors was obtained from the AGRIS database (https://agris-knowledgebase.org/AtTFDB/, accessed on 15 April 2021).

### 2.4. Protein-Protein Interactions and Network Construction

A network analysis on protein–protein interactions (PPI) was performed to uncover any plausible interactions among proteins for which the DEGs were found to be significantly different. TF genes and genes in the literature review that overlapped with significantly differentiated DEGs were considered for the PPI analysis. The STRING database [39] was employed to enable the PPI network analysis. Cytoscape software was used for visualizing the interaction networks.

### 2.5. Prediction of Potential miRNAs

MicroRNAs associated with plant diseases must be identified before plant–pathogen interactions can be studied. This identification is also necessary to understand how plants defensively respond to pathogens. Identifying potential and small RNAs is possible using the psRNATarget server, wherein parameters are set to a default, except in the case of maximum expectation, which was set to 2 for this study. psRNATarget evaluates complementarity between small RNA and target gene transcripts using a scoring scheme originally implemented by miRU. psRNATarget uses the popular Smith-Waterman implementation of SSEARCH (version 36.x), because it is guaranteed to find the most alignments between small very short RNA sequences and mRNA sequences (http://plantgrn.noble.org/psRNATarget/, accessed on 1 May 2021).

## 3. Results and Discussion

### 3.1. Identification of Differentially Expressed Genes

To identify host genes responsive to pathogens that commonly infect *Arabidopsis*, RNA-Seq data were retrieved from seven independent experiments, consisting of 181 samples of infected plants that had been infected either by fungi, oomycete, bacteria, or viruses. Another set of RNA-Seq data were also retrieved from 103 samples of mock-inoculated plants (Table 1). Of these, 3694 DEGs were significantly different from the healthy control plants, with 1909 up-regulated and 1785 down-regulated DEGs (Appendix A). Not surprisingly, the data overrepresented defense-related genes that are already known in plants. These genes include *PR12* (*AT1G75830*), *FMO1* (*AT1G19250*), *LECRK-I.1* (*AT3G45330*), *GLIP1* (*AT5G40990*), *WRKY75* (*AT5G13080*), and *WRKY51* (*AT5G64810*), all of which were highly induced [40]. An enhanced defense-related gene, *FMO1*, involved in basal resistance against virulent pathogens was identified. It contributes to the regulation of *EDS1* and induces resistance by causing the death of plant cells at pathogenically infected sites [41,42]. The membrane-spanning receptor-like kinase, *LECRK-I.1*, is required for environmental stress responses [43]. Lectin receptor-like kinases belong to a specific PRR group that perceives PAMPs and initiates defense responses [44,45]. *GLP 1* regulates plant immunity by regulating ethylene signaling [46,47] and by members of the *WRKY* transcription factor gene family (e.g., *WRKY75* and *WRKY51*). These two are known to be involved in prompting the plant response to oxidative stress and ROS homeostasis, respectively [48,49,50], whereas *WRKY-75* and -*51* are involved in defense responses that are possibly induced by jasmonic acid.

In contrast with defense-related genes, those involved in cellular and metabolic processes, such as photosynthesis, were generally down-regulated in this study. In particular, this was evident in genes that regulate photosystem-II; i.e., *PSBP-2* (*AT2G30790*), *DEG13* (*AT3G27690*), and *PsbP-2* (*AT2G30790*) [51,52].

### 3.2. Gene Ontology Confirmed Strong Impact on Diverse Cellular Processes

To investigate the functions of the DEGs, a GO was performed to explore the plant–pathogen interaction. There were 59 genes related to cellular components whose expression was significantly altered after infection, 26 of which were up-regulated and 33 of which were down-regulated. Expression was also significantly altered in 23 genes that shape molecular functions, 13 of which were up-regulated and 10 of which were down-regulated. In addition, there were notable alterations in the expression of 60 genes related to biological processes, 42 of which were up-regulated and 18 of which were down-regulated (Figure 2, Appendix A). The up-regulated genes were largely involved in oxidation-reduction processes (GO: 0055114), protein phosphorylation (GO: 0006468), and defense responses (GO: 0006952); down-regulation also occurred in genes that code for oxidation-reduction processes (GO: 0055114), as well as photosynthesis (GO: 0015979) and response to cold (GO: 0009409) (Figure 2a,b). Oxidation-reduction signaling acts as a general plant response to most pathogens [53]. Some genes associated with reduction-oxidation (redox) processes, such as NADPH oxidases and catalases, are required for immunity, suggesting that the redox state might add an additional layer of regulation to plant defense responses [54]. Thus, our findings verify the hypothesis that multiple ROS signals are integrated together during a defense response [55].

Furthermore, plant molecular functions including ATP binding (GO: 0005524), kinase activity (GO: 0016301), and protein serine/threonine kinase activity (GO: 0004674) were up-regulated, whereas oxidoreductase (GO: 001649), catalytic (GO: 0003824), and rRNA binding (GO: 0019843) activities were down-regulated (Figure 2c,d). In support of molecular functions and biological processes, up-regulation occurred in DEGs responsible for the integral component of cellular membranes (GO: 0016021), the plasma membrane (GO: 0005886), and cytoplasm (GO: 0005737), whereas down-regulation was observed in the case of integral component of membranes (GO: 0016021), the plasma membrane (GO: 0005886), and chloroplasts (GO: 0009507) (Figure 2e,f).

The plasma membrane in plant cells is a main barrier against pathogenic attack, as it mediates the interactive relationship between plant cells and pathogens. In the first step, pathogens are recognized by plant cells at the plasma membrane. Many of the initial cellular reactions to pathogenic infections are coordinated by ion channels and plasma membrane-localized enzymes [56]. As a result, multiple downstream responses to pathogenic invasions are expressed in the plasma membrane. For pathogens, finding access to plant cells and their nutrients requires the manipulation of host cells in a manner that would suppress their defensive responses. Accordingly, the membrane is expected to serve multiple functions in plant–pathogen interactions [57,58]. The recognition of PAMPs (pathogen-associated molecular pattern) is mediated by recognition receptors in plasma membrane-localized patterns, often belonging to a class of enzymes known as Receptor-Like protein Kinases [*RLKs*], such as *CERK1* [56]. Upon recognizing PAMPs, RLKs quickly activate their co-receptors (i.e., *BAK1*, *SERK4*, *BIR2*, *CRK28*, *IOS1*, *PSKR1*, *ERECTA*, or *RLP51*) as the first step in intracellular signaling, thereby regulating cellular activity even as the pathogen infection prompts the release of cell wall components, antimicrobial compounds, and defense-related proteins [56,59,60,61]. The GO analysis revealed that similar mechanisms in functional responses are induced, even though the analyses explored the differential responses of plants to different pathogens (Figure 2).

### 3.3. Enrichment Analysis of the KEGG Pathway Highlights the Reticulate Nature of Defense Metabolism in Plants

KEGG pathway enrichment analyses were performed on DEGs with *p*-values lower than 0.05. It appeared that 11 pathways are significantly up-regulated and 18 are down-regulated in response to infection (Appendix A). The enriched metabolic pathways included those responsible for the biosynthesis of secondary metabolites and amino acids, along with genes involved in photosynthesis and plant–pathogen interactions (Table 2). The metabolic signature of the KEGG pathway analyses highlights the reticulate nature of the metabolic plant responses, characterized by crosstalk among hormones and features of the biosynthesis of specialized products, in addition to general changes in metabolism and plant function [62,63,64]. Pathogen or pathogen-derived elicitors alter the metabolism of carbohydrates, amino acids, and lipids [65,66,67,68], with amino acids and sugars being intermediates of the pathways that synthesize specialized defense metabolites [1,11,69].

For instance, the biosynthesis of amino acids occurs by previously identified genes that encode arogenate dehydratase 4 (*ADT4*), anthranilate synthase beta subunit 1 (*ASB1*), tryptophan synthase beta-subunit 1 (*TSB1*), tryptophan synthase alpha chain (*TSA1*), and tyrosine aminotransferase 3 (*TAT3*). These amino acids were found to be collectively involved in plant responses to stress. Plant immune responses against pathogens can include the biosynthesis of bioactive molecules in plants, most of which have antimicrobial effects that are either ‘phytoanticipins’ and/or ‘phytoalexins’ [70,71]. In this regard, 239 genes were identified in the biosynthesis of the secondary metabolite pathway as the second largest DEG pathway. The synthesis of secondary metabolites is a defense mechanism in response to phytopathogens. Secondary metabolites, including terpenes, phenolics, nitrogen (N), and sulphur (S) containing compounds act as chemical barriers that protect plants against biotic and abiotic stresses [72]. Antimicrobial secondary metabolites are classified into phytoalexins and phytoanticipins [73]. Phytoalexins are a defensive compound and a secondary metabolite induced by the hypersensitive response during plant–pathogen infection [74]. These defense-related compounds are based on the prominent biochemical capacity of a plant. Many plants’ defensive arsenals are taken from amino acid precursors, for example glucosinolates products, which are pivotal in defensive responses of plants against pathogenic invasion [71,75,76]. Previous studies demonstrate that the metabolism of amino acids and peptides is essential for the biosynthesis of many natural substances that protect plants from pathogenic invasion. As a result, they promote the plant immune system [77,78,79]. However, a series of studies on *Arabidopsis thaliana* concluded that the significance of amino acid metabolism, when playing a role in plant–pathogen interactions, goes far beyond functions in secondary metabolite production [80,81].

In this study, 29 genes were found to be involved in photosynthesis, including *FNR1* (*AT5G66190*), *PETC* (*AT4G03280*), *FNR2* (*AT1G20020*), *PDE334* (*AT4G32260*), and *ATPD* (*AT4G09650*). The expression level of these genes was affected in response to the presence of pathogens. Several studies have shown the suppression of photosynthesis in infected plants, perhaps reflecting active responses by plants that restrict carbon availability, and thus limit the growth of pathogen(s) in the plant. Photosynthetic functions may also be suppressed because plants favor the establishment of defense over other physiological processes at times of pathogenic attack [82,83]. Photosynthesis takes place in the chloroplasts and generates important substances such as carbohydrates, ATP, and *NADPH*, which are utilized in several biosynthetic pathways that produce amino acids, hormones, and secondary metabolites. They are also instrumental for the success of immune responses and for cells to identify environmental stress signals. Chloroplasts are basic generators of ROS and NO, which are pivotal for defensive barriers in plants. Defense-related signaling molecules and hormones may also influence photosynthesis. Exogenous treatments of plants with these substances reduced photosynthetic pigments and tended to cause stomatal closure [84,85]. Notably, salicylic acid (SA), ethylene (ET), and ROS can have both positive and negative effects on photosynthetic function [86,87]. Furthermore, excess quantities of ROS can damage photosynthetic complexes, especially *PSII*, thereby instigating photo inhibition [88,89,90,91].

### 3.4. Identification of Transcription Factors

The capacity of transcription factors is a key tool for the regulation of plant responses to stress. It is therefore essential to identify genes that encode transcription factors. In the current study, we found that transcription factors belonging to the WRKY-family were mostly induced by pathogens. Up-regulation was observed in Alfin-like, CCAAT-HAP3, E2F-DP, RAV, VOZ-9, HSF, and NAC, whereas down-regulation occurred in ARF, ARID, ARR-B, C2C2-CO-like, C2C2-YABBY, CCAAT-HAP2 G2-like, GeBP, GRAS, NLP, PHD, SBP, and ZF-HD families (Figure 3). Alfin-like, ARID, CCAAT-HAP2, CCAAT-HAP3, E2F-DP, G2-like, GeBP, RAV, and VOZ-9 all belong to single-gene families represented by only one gene per family (Figure 3). Further information about the transcription factors of genes is given in Appendix A. Transcription factors-encoding genes may be differentially up-regulated or down-regulated under stress conditions. As such, there is a significant overlap in immune response pathways which allows different defense signals to be integrated. This process is expected to promote plant defense against pathogenic attack [92,93]. TFs involved in the defense pathways and responses against pathogens mostly belong to the WRKY and NAC families. The relation between these families might have a positive role in plant resistance and could increase plant immune responses against pathogens via regulating defense gene expression.

### 3.5. Protein–Protein Interaction of Transcription Factors and Hub Genes Identification

To better understand the diverse landscape of defense-related activated genes, PPI networks were constructed based on the DEGs results. The resulting PPI network revealed 199 nodes and 579 edges (Figure 4); the most significant nodes (hubs) were *WRKY40*, *WRKY46*, *STZ*, *WRKY18*, *NPR1*, *RHL41*, *WRKY70*, *WRKY25*, *WRKY53*, *EDS1*, *MYB15*, *MPK11*, *SARD1*, *AT5G66070*, *PDF1.2*, *TIP*, *NAC062*, *BZIP60*, *FRK1*, and *SCL13*. This set of proteins includes proteins that are directly or indirectly regulated by pathogens and are differentially expressed when cells are infected by pathogens. Some of the proteins associated with the plasma membrane play important roles in the immune response to pathogens, including *NAC062* and *E3 ubiquitin ligase* (AT5G66070). E3 Ub-ligases are associated with the regulation of cellular perception of pathogens by PRRs at the plasma membrane, and also having a role in defense-related signaling. E3 ubiquitin-ligases regulate cellular perception of pathogens when the innate immunity in plants functions normally [94].

The WRKY family members also play a pivotal role in plant defensive responses [95]. *WRKY40* and *WRKY18* play a role in pathogen-induced HR, in association with the induction of SA-mediated immune responses that contribute to systemic acquired resistance. These WRKYs negatively affect resistance to hemibiotrophic pathogens [96]. *WRKY53* is a component of the systemic acquired resistance signaling network [97,98] that interacts with the *ESR* and inducible proteins involved in jasmonic acid-related responses to mediate the negative crosstalk between senescence and pathogen resistance [99]. Genetic analyses suggest that *WRKY53* is able to regulate the expression of *WRKY46* and *WRKY70* in basal resistance [100]. *WRKY25* is an essential regulator of SA-mediated immune responses [101,102], whereas *WRKY70* converges between signals of SA and JA in plant defense responses [103]. NONEXPRESSOR PATHOGENESIS-RELATED GENES1 (NPR1) is a hub regulator of defense responses mediated by SA [104], and involved in defense layers of PTI and ETI. It is also a fundamental molecular regulator for the programing of cell death in ETI. *EDS1* is a positive switch of basal resistance, which is needed for ETI [105].

### 3.6. Identification of Potential miRNAs

MicroRNAs (miRNAs) are a class of short endogenous non-coding small RNA molecules. Many of these molecules play substantial roles in the defense response at times of biotic stress. To find putative miRNAs associated with DEGs, we utilized the computational algorithm psRNATarget with high specificity based on the choice of penalty score (≤2 being highly stringent). We detected 180 miRNAs, of which 39 belonged to conserved families (Figure 5). The miR5021 family possesses the highest frequency (39 members), followed by miR156, miR5658, and miR414 (20 members each), in addition to several members having already-known stress response associations (i.e., miR169, miR395, miR399, miR393, miR156, miR171, miR172, miR161, miR163, and miR165) as regulators of PTI and ETI [106].

## 4. Conclusions

The analysis of transcriptomic datasets available for similar plant host responses to different biotic stressors provided a unique opportunity to obtain critical knowledge about genes that are commonly expressed during pathogen infection, and will be useful in the development of novel genetic resistance strategies. Considering the inevitable variation between different studies and their considerable effects in the type and number of transcripts—owing to variations in host plant ecotype, pathogen species or isolate, and experimental conditions—we integrated the available transcriptomic datasets of *Arabidopsis thaliana* during responses to several pathogens in a coherent approach that identified commonalities pertaining to pathogenic stress responses in other host plants. Transcriptomic analysis and the PPI network analysis helped researchers to visualize and introduced key node proteins that counteracted pathogenic infections. More importantly, they could serve as new biomarkers in diagnosing and guiding management strategies of plant disease. This study highlights the importance of TFs as regulators of plant responses against pathogens. Future research can be directed at candidate genes as general indicators of resistance and encompass different mechanisms of defense against a broad range of pathogens in *Arabidopsis*. Further studies are required to elucidate the molecular mechanisms of DEGs, TFs, and miRNAs, and uncover the relation between them in plant disease resistance against pathogens.

## Figures and Tables

**Figure 1 biology-11-01155-f001:**
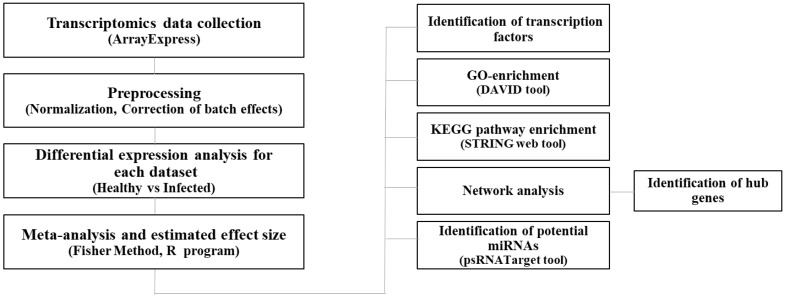
Schematic overview of the integrative strategy for understanding aspects of common responses of *Arabidopsis* to various pathogens.

**Figure 2 biology-11-01155-f002:**
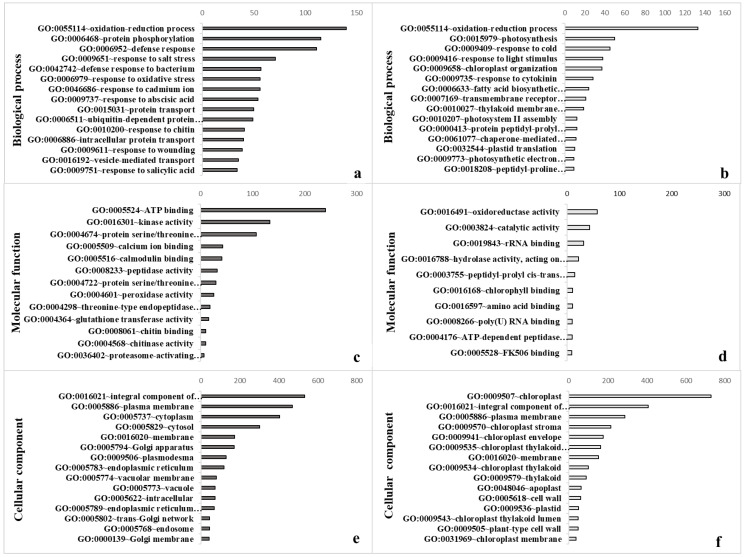
Gene ontology enrichment analysis of the DEGs. The enriched genes were sorted into three categories according to gene function: (**a**,**b**) biological process (e.g., active in defense responses and photosynthesis), (**c**,**d**) genes involved in molecular functions (redox and energy metabolism), and (**e**,**f**) genes responsible for synthesis and organization of cellular components (e.g., with importance for membrane and organelle structures). Up-regulated genes are listed on the left panel (dark grey) and down-regulated genes on the right (light grey).

**Figure 3 biology-11-01155-f003:**
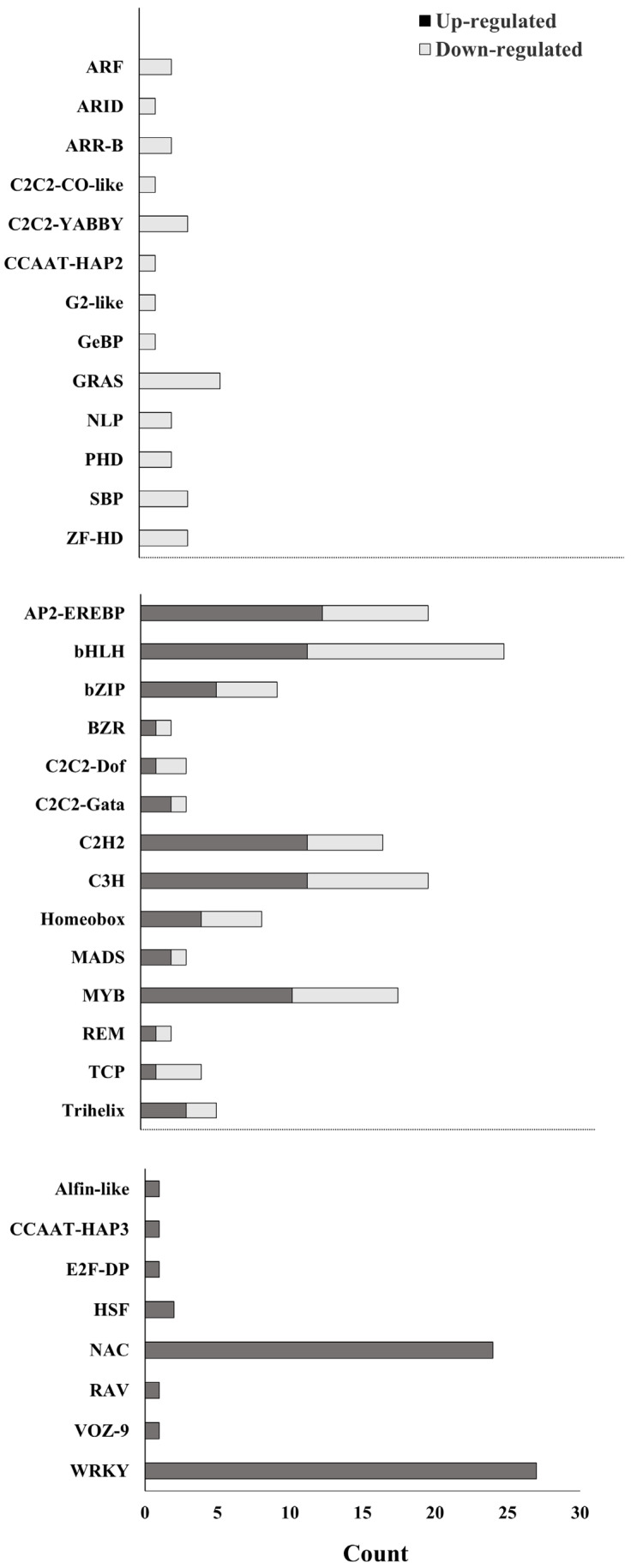
Transcription factors with significant responses to pathogen treatment, indicating direction of change in gene activity: up-regulated (dark grey); down-regulated (light grey).

**Figure 4 biology-11-01155-f004:**
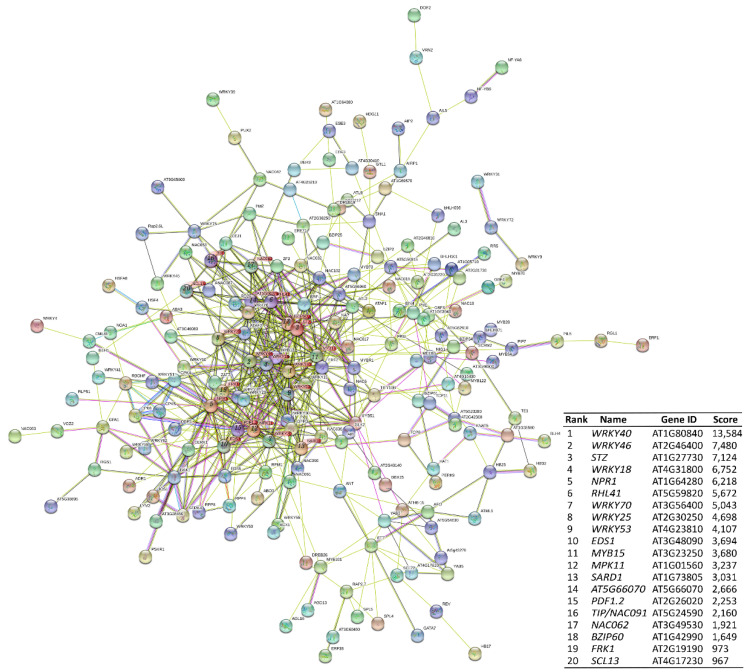
Protein–protein interaction network highlighting hub genes involved in plant–pathogen interaction in *Arabidopsis*. The most important hubs are ranked based on their importance in the network.

**Figure 5 biology-11-01155-f005:**
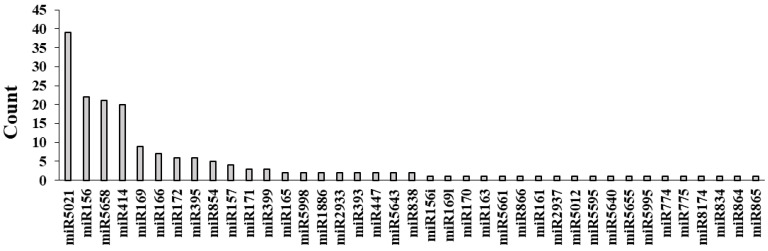
miRNAs associated with the DEGs and discovered by use of the computational algorithm psRNATarget server. The filtering was performed based on a highly stringent penalty score (≤2).

**Table 1 biology-11-01155-t001:** Transcriptomic raw data related to plant–pathogen interaction studies of *Arabidopsis thaliana* used for the current meta-analysis.

Accession Number	Pathogen Species	Samples Number	Control Number	Plant Part	Related Article
E-MTAB-4151	*Pseudomonas syringae* pv. *maculicola*	12	12	Leaf	[25]
E-GEOD-53641	*Hyaloperonospora arabidopsidis*	144	72	Aerial shoots	[24]
E-GEOD-34241	*Fusarium oxysporum*	4	4	Whole plants	[29]
E-MTAB-4416	*Pseudomonas syringae*	3	3	Leaf	[26]
E-GEOD-56922	*Cabbage leaf curl virus*	4	4	Leaf	[30]
E-MTAB-4281	*Botrytis cinerea*	2	2	Whole plants	[28]
E-MTAB-4450	*Pseudomonas syringae*	12	6	Leaf	[27]

EBI The European Bioinformatics Institute.

**Table 2 biology-11-01155-t002:** The KEGG pathway enrichment of the total of differentially expressed genes (DEGs).

Pathway	Gene Count	Adjusted *p* Value
Metabolic pathways	382	0.000010
Biosynthesis of secondary metabolites	239	0.000000
Carbon metabolism	76	0.000006
Biosynthesis of amino acids	74	0.000006
Plant-pathogen interaction	47	0.000293
Proteasome	37	0.000000
Glutathione metabolism	32	0.000192
Glycolysis/Gluconeogenesis	32	0.006589
Photosynthesis	29	0.000070
Glycine, serine and threonine metabolism	25	0.000993
2-Oxocarboxylic acid metabolism	25	0.001530
Glyoxylate and dicarboxylate metabolism	25	0.001530
Phenylalanine, tyrosine and tryptophan biosynthesis	20	0.003251
Pentose phosphate pathway	19	0.004130
Arginine biosynthesis	14	0.005035

## Data Availability

The data presented in this study are available in the Appendix A.

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
