# Peer review of "Transcriptome Meta-Analysis Identifies Candidate Hub Genes and Pathways of Pathogen Stress Responses in Arabidopsis thaliana"

_biology, 2022, doi:10.3390/biology11081155_

Round 1

Reviewer 1 Report

The manuscript "A meta-analysis for investigating cross-talk among transcriptional response to biotic stresses in Arabidopsis thaliana" by Biniaz et al deals with a relatively small but very stringent analysis of already published next generation sequencing data in Arabidopsis thaliana under different pathogen infections.  Among the objectives of the study, the uncovering of common pathways and genes up or down-regulated by all the assessed pathogens was paramount.

The study is justified by the lack of systematic comparison among the datasets from different pathogens. The authors are right and this approach is welcome. Taking advantage of all the genomic features Arabidopsis provides, the researchers came up with a subset of transcriptional studies that allow for a stringent and in depth analysis.

Although the authors produced very interesting data, this reviewer has several suggestions and questions that may help produce an even better manuscript.

The authors included fungi, oomycete, bacteria and a virus (7 pathosystems in total). However, why not include nematodes, RNA viruses (only a ssDNA virus was included in the study), and other biotic stresses for which there are transcriptional analysis of quality? (even not all of them by next generation sequencing methods). An example of this would be the works of Whitham et al (2003 and 2006) that although done with microarrays, it is validated and provides with a global approach as this works is attempting.

Furthermore, this reviewer consider that the authors missed a grand opportunity of also highlighting the differences between the responses to  these pathogenic organisms and how Arabidopsis induces or represses groups of genes, pathways or transcription factors differently, depending on the type of pathogen. This would be a relatively simple analysis to do by the authors that may produce a novel understanding of the host response.

Please find some suggestions and corrections that were found through the manuscript:

2. Results and discussion:

The first sub-header is numbered 1.2, it should be 2.2, and the rest are also wrongly numbered in Results.

Lane 159, it says... "to identify genes of pathogens that are commonly involved in infecting Arabidopsis".. it should probably say ..."to identify host genes responsive to pathogens that are...."

Lane 239, why not place the Table S2 in the main body?, it seems those are really relevant for the manuscript.

Lane 293, It says " Transcription factors that encode genes may be differentially..." it may say " Transcription factors-encoding genes may be differentially..."

I noticed that through the manuscript, sometimes the authors name a gene but also include the Arabidopsis Gene Identifier (example in Lane 168). This is really appreciated, however, is not done through the whole manuscript. Please do so or, as a suggestion, make a list including all the names of the genes found in the manuscript with their corresponding AGI number, it will be very useful for the reader, especially since you mention a lot of genes.

Figure 4 is very difficult to read at its current size.

Lane 366 shows the word "essential" at a different font size than the rest, is this intentional?

Lane 419, ", we identified 220 genes... maybe would be better to describe what it means, like... we identified 220 transcripts that are regulated in the pathogenic state (or something like that). It is left very open.

And for the title, may it be better something like "A meta-analysis identifies response hubs among biotic stress transcriptional responses in Arabidopsis thaliana", just and idea.

Author Response

Our thankful to the reviewers and the editor for their thoughtful critiques of our manuscript. We think that the manuscript has been greatly improved after this revision. We responded all of the comments/suggestions. All the changes are track changed in the manuscript text and our point-by-point responses to the reviewers’ comments were mentioned. "Please see the attachment."

Reviewer 2 Report

In the current study under review, the authors made an attempt to examine the changes in transcriptome profiles of Arabidopsis thaliana in response to different biotic stresses. However, some major changes need to be made to increase the readability and enhance the presentation of the obtained results.  Furthermore, some queries need to be answered/clarified before the manuscript could be judged as suitable for publication. The following are some comments, suggestions, or queries regarding the manuscript.

1.     Please check the attached PDF file for specific comments.

2.     Abstract is very lengthy. Lines 17-20 and 34-35 could be removed.

3.     Introduction is sufficient.

4.     Materials and methods need major changes as follows:

a.      Figure 1 shows a schematic overview of the analysis performed; however, no information regarding the tools used for data “cleaning” and alignment against the reference genome is shown. Without this information, the analysis is not reproducible.

b.     Which genome version was used as reference and from where was it obtained?

c.      What was the criteria (fold change) used for identifying the DEGs?

d.     More information regarding the PCR enrichment is needed. At least, the authors should provide information about the kits used for this purpose.

e.      How many experiments were included in the analysis (7 or 8)? Also how many samples? Numbers are not the same in methods and results?

f.      How were the transcription factors identified?

5.     Results

a.      The authors explained that some genes were up- or down-regulated; however, the |fold change| of these genes doesn’t exceed 0.5 which could not be by any means considered as change in regulation without verification (in vivo) using, for example, qRT-PCR.

b.     Were the up- and down-regulated genes combined for GO and pathway analysis or used separately?

c.      Figure 2 quality is so bad. Moreover, the authors explained that “59 genes related to cellular components whose expression was significantly altered after infection, 26 of which were up-regulated and 33 were down-regulated”. What does this sentence mean? The number of genes should be shown in each category or do they mean the number of categories. But the figure shows only 15 up- regulated and 15 down-regulated categories. It is really very confusing and applies to the whole part of GO analysis. This section needs to be re-written.

d.     The numbers of genes in each enriched KEGG pathway are not the same in Table 2 and Table S2. What is the difference between them and why did the authors show these results twice?

e.      The authors explained the roles of different TF families without showing the relation between them and the results obtained in their current study.

f.      Similarly, the authors explained the roles of different miRNA families without showing the relation between them and the results obtained in their current study

g.     The quality of all figures is not acceptable.

h.     Numbering of headings is not correct and largely misleading.

6.     Conclusions are merely a repetition of what was found in the results. The authors should show here their insights about the obtained results. What are the advantages of their study? What are the factors that hinder the analysis including, for example, the lack of information regarding the variation in growth conditions applied in each study? What are the future applications of the obtained study and the planned future research?

7.     Overall, a significant improvement in the manuscript preparation and the implementation of the experiment is needed.

Author Response

(The authors gave the same response as above.)

Reviewer 3 Report

This work describes an integration of the accessible transcriptomic data regarding general molecular plant defense responses to different pathogen attacks in Arabidopsis thaliana, summing up the results of several studies by using a novel approach. Since the information of the molecular basis of this relations are rather limiting, exploiting meta-analysis as powerful means can be highly beneficial enabling the detection of essential genes and signaling pathways involved in regulation of complex processes in plants’ defense systems.

Overall, the manuscript is very well written, with complete and corresponding Discussion section, leading to conclusions highlighting the significance of the accomplished results. It represents a convenient follow up or continuation of authors’ recently published paper “Meta-Analysis of Common and Differential Transcriptomic Responses to Biotic and Abiotic Stresses in Arabidopsis thaliana“, in Plants (MDPI). Even though that these papers emphasize on different groups of genes and pathways, one must notice that the form and the core of both papers are quite similar, practically encompassing the same data.

As for reviewed manuscript itself I have only few minor remarks concerning the Material and Methods section, and the use of abbreviations and full names of the represented genes, as I pointed out in my comments in PDF document.

Author Response

(The authors gave the same response as above.)

Reviewer 4 Report

The manuscript reports a meta-analysis of transcriptomic data in plant defense responses to pathogen attacks in Arabidopsis. Analysis includes DEGs, Gene Ontology, KEGG Enrichment pathways, transcription factors, miRNAs and protein-protein interaction networks. Overall, the results should be of value to the field of plant responses to the biotic stress. However, there are some issues need to be addressed.

Major issues.

1. Data Collection. Line 96- 97, “These RNA-seq were obtained from the ArrayExpress of the European Molecular Biology Laboratory–European Bioinformatics Institute (http://www.ebi.ac.uk/arrayexpress).” 

Why did authors only choose this database as the expression (RNA-seq) data collection? There are some important RNA-seq databases have more abundant expression data, especially in NCBI (National Center for Biotechnology Information). Why did not choose NCBI?

 2. The title of the paper is “…biotic stresses…”, however, biotic stresses are made by not only bacteria, fungi, and viruses, but also animals, such as insect herbivory. The authors should add the insect herbivory related analysis to the results.

3. The KEGG pathway enrichment of the total differentially expressed genes (DEGs).

The results of Table 2 show that the second largest DEG pathway is “Biosynthesis of secondary metabolites, Gene Count 239”. However, the authors did not describe any enrichment pathway of “Biosynthesis of secondary metabolites” in the results. Secondary metabolite synthesis is very important for plant defense and immunity to the biotic attack (Int J Mol Sci, 2022, 23, 7031). The authors should add secondary metabolites as the defense compounds to biotic stress in KEGG pathway enrichment analysis.

Minors:

1. Line 166-167, “AT1G75830, AT1G19250, AT3G45330, AT5G40990, AT5G13080,AT5G64810 and AT1G75830” What are the names of these genes? Same question is in line 182, “AT2G30790, AT3G27690…. AT3G27690”.

2. Most of paragraphs are numbered wrongly. Such as line 94,112,127 are all 1.1;  line143,150, 159, 184, 283 are all 1.2…………………Please check it and renumbered carefully.

3. Keywords: Change “plant response to multiple biotic stresses” to “biotic stress”.

4. All gene symbols should be italicized.

5. References. Some journal names are upper while others lower case. The format of references should be unified.

Author Response

(The authors gave the same response as above.)

Round 2

Reviewer 2 Report

Please provide your response for the comments raised in the first round. 

Some questions were not answered:

1. What was the criteria (fold change) used for identifying the DEGs?

2. How were the transcription factors identified?

3. The authors explained that some genes were up- or down-regulated; however, the |fold change| of these genes doesn’t exceed 0.5 which could not be by any means considered as change in regulation without verification (in vivo) using, for example, qRT-PCR.

4. Were the up- and down-regulated genes combined for GO and pathway analysis or used separately?

5. Figure 2 quality is so bad. Moreover, the authors explained that “59 genes related to cellular components whose expression was significantly altered after infection, 26 of which were up-regulated and 33 were down-regulated”. What does this sentence mean? The number of genes should be shown in each category or do they mean the number of categories. But the figure shows only 15 up- regulated and 15 down-regulated categories. It is really very confusing and applies to the whole part of GO analysis. This section needs to be re-written.

6. The authors explained the roles of different TF families without showing the relation between them and the results obtained in their current study. Similarly, the authors explained the roles of different miRNA families without showing the relation between them and the results obtained in their current study. These parts should be removed or the authors should have to show their relation to the study.

7. Still the quality of all figures is not acceptable.

Round 3

Reviewer 2 Report

The authors made all the required corrections.